# Infectious Keratitis: A Tertiary Center’s Approach to Diagnosis, Management, and Enhanced Outcomes Through Microbiological Analysis

**DOI:** 10.3390/microorganisms13102308

**Published:** 2025-10-05

**Authors:** Antonio Moramarco, Federico Cassini, Natalie di Geronimo, Giovanni Zanini, Michele Potenza, Matteo Farnè, Viviana Schisa, Erica De Carolis, Margherita Ortalli, Piera Versura, Tiziana Lazzarotto, Luigi Fontana

**Affiliations:** 1Ophthalmology Unit, Dipartimento di Scienze Mediche e Chirurgiche, Alma Mater Studiorum, University of Bologna, 40126 Bologna, Italy; federico.cassini@studio.unibo.it (F.C.); giovanni.zanini@studio.unibo.it (G.Z.); michele.potenza2@studio.unibo.it (M.P.); piera.versura@unibo.it (P.V.); luifonta@gmail.com (L.F.); 2IRCCS Azienda Ospedaliero-Universitaria di Bologna, 40138 Bologna, Italy; natalie.digeronimo@icloud.com; 3Department of Statistical Sciences “Paolo Fortunati”, Alma Mater Studiorum, University of Bologna, 40126 Bologna, Italy; matteo.farne@unibo.it (M.F.); viviana.schisa2@unibo.it (V.S.); erica.decarolis@studio.unibo.it (E.D.C.); 4Department of Medical and Surgical Sciences, Alma Mater Studiorum, University of Bologna, 40138 Bologna, Italy; margherita.ortalli2@unibo.it (M.O.); tiziana.lazzarotto@unibo.it (T.L.); 5Microbiology Unit, IRCCS Azienda Ospedaliero-Universitaria di Bologna, 40138 Bologna, Italy

**Keywords:** corneal infections, microbial keratitis, bacterial keratitis, fungal keratitis, viral keratitis, *Acanthamoeba keratitis*, epidemiology, ophthalmology

## Abstract

Background: The study aimed to assess the diagnostic and therapeutic management of infectious keratitis at a tertiary referral center, focusing on how microbiological analysis influences clinical outcomes. Methods: A retrospective review was conducted on 220 patients (221 eyes) with infectious keratitis treated between November 2021 and January 2025. Data collected included clinical presentation, microbiological findings, treatment approaches, and outcomes. Statistical analyses examined the relationships between microbiological results, improvements in visual acuity, and the need for rescue surgery. Results: Bacterial keratitis accounted for 64% of cases, followed by viral (20%), fungal (13%), and Acanthamoeba (3%). Microbiological testing was performed in 107 cases, with a culture positivity rate of 75.7%. Positive microbiological findings were significantly associated with better visual acuity (*p* = 0.019) and a reduced, though not statistically significant, need for rescue surgery. Use of contact lenses and ocular trauma were independent risk factors for culture positivity. Delayed referral (more than 15 days) was linked to longer treatment durations and a higher likelihood of surgical intervention (*p* < 0.001). Microbiological diagnosis correlates with improved visual outcomes and a decreased need for surgical procedures. Conclusion: Early referral and targeted therapy are essential for optimizing prognosis. The use of contact lenses and cases of ocular trauma should prompt early diagnostic sampling.

## 1. Introduction

Microbial keratitis (MK) is an infectious disease of the cornea characterized by ocular pain, conjunctival hyperemia, stromal infiltration, and often corneal ulceration with subsequent vision loss. Globally, MK is the fourth leading cause of blindness, with an estimated 1.5 to 2.0 million new cases reported annually in developing regions [1,2,3]. In contrast, the incidence in developed countries remains considerably lower, ranging from 3.6 to 40.3 cases per 100,000 person-years [4,5,6,7]. Due to its potential to cause rapid and permanent visual loss, MK is regarded as a major ophthalmic emergency and a significant public health concern. Prompt microbiological diagnosis and early targeted therapy are vital to prevent corneal perforation and severe visual impairment. The range of causative microorganisms—including bacteria, fungi, protozoa, and viruses—varies according to geographic, climatic, socioeconomic, and host-related risk factors [8,9,10].

Bacterial keratitis (BK) accounts for about 90% of all MK cases [11]. Major risk factors include contact lens use, ocular trauma, topical corticosteroid use, and a history of previous ocular surgeries [12]. Fungal keratitis (FK) has a worldwide distribution, with causative agents heavily influenced by geographic location. In tropical and subtropical regions, filamentous fungi are most common, while yeast species are more prevalent in temperate areas. *Acanthamoeba keratitis* (AK) is a rare but increasingly recognized cause of MK, caused by free-living amoebae found widely in natural environments [13]. Over the past decade, AK cases have risen, mainly linked to contact lens wear, which accounts for about 90% of reported cases [14]. Lastly, viral keratitis (VK) is among the most common types of infectious keratitis [15,16]. Of the various viruses causing keratitis, the alpha-herpesvirus herpes simplex virus (HSV) is the most common. Other common viral agents include the beta-herpesvirus cytomegalovirus (CMV), the alpha-herpesvirus varicella-zoster virus (VZV), and the gamma-herpesvirus Epstein–Barr virus (EBV) [17,18,19].

The main aim of our study was to describe the experience of our third-level center in diagnosing and treating infectious keratitis. Particular emphasis was placed on the use of microbiological analysis to identify the microorganism responsible for the infection. Secondary objectives included identifying potential risk factors associated with a higher likelihood of obtaining a positive microbiological result, and comparing the two groups of patients—those with positive and negative microbiological findings—in terms of treatment strategies and clinical outcomes. This analysis may provide insights to help prevent, enable early intervention, and improve the management of infectious keratitis in specialized care settings.

## 2. Materials and Methods

This study involved 220 patients who presented to the Cornea and Ocular Surface Service at our tertiary referral ophthalmology center over a three-year period (November 2021–January 2025). Patients who had not concluded their therapeutic course yet or were still under clinical surveillance by the end of January 2025 were excluded from the cohort. The study adhered to the principles outlined in the Declaration of Helsinki. Ethical approval was granted by the Institutional Review Board (Registration No. 901/2022/Oss/AOUBo, dated 15 December 2022) of IRCCS AOU Policlinico Sant’Orsola, and informed consent was obtained from all participants prior to enrolment. Clinical data were collected and systematically categorized according to five key stages of the diagnostic-therapeutic process: anamnesis, clinical presentation, microbiological analysis, treatment, and outcomes.

Microbiological samples were collected using various methods, including corneal scraping, conjunctival and corneal swabbing, as well as the collection of contact lenses and their storage media. The Microbiology Laboratory carried out cultures on different media types, such as standard agar, blood agar, Sabouraud agar, and enriched broth media. In some cases, and in every suspect of amoebic keratitis, non-nutrient agar (NNA) overlaid with *Escherichia coli* and polymerase chain reaction (PCR) assays were also utilized to enhance pathogen detection.

### Statistical Analysis

Statistical analyses were performed to investigate the relationship between microbiological findings, treatment strategies, and visual outcomes. All variables were processed based on their measurement scale and clinical significance. BCVA (best corrected visual acuity) improvement was determined as the absolute difference between final and baseline visual acuity, expressed in decimal units.

Welch’s *t*-test was used to compare the mean BCVA improvement between patients with positive and negative microbiological results, as it is resilient to violations of the equal-variance assumption [20]. Distributions of continuous outcomes across categorical groups were further explored using boxplots, which provided insights into patterns of BCVA and referral latency (defined as the number of days between symptom onset and presentation at the specialized center) by microbiological status, as well as differences in visual outcomes based on referral timing. The linear relationship between continuous variables—specifically, referral latency and BCVA improvement—was assessed using Pearson’s correlation coefficient. Associations between categorical variables were examined using Fisher’s exact test. These included the relationship between microbiological findings and the performance of rescue surgery, as well as the association between referral latency and both visual improvement and surgical intervention, all treated as dichotomous.

Logistic regression models [21] were fitted to evaluate two distinct outcomes: first, among patients who underwent microbiological testing, to identify risk factors linked to a positive culture result; and second, whether referral latency was associated with the likelihood of undergoing rescue surgery. Variable selection was performed using a backwards elimination strategy based on the Least Absolute Shrinkage and Selection Operator (LASSO), retaining only predictors with a significance level below 5% [22,23].

Furthermore, patients’ medical histories were examined to identify potential risk factors linked to a positive microbiological outcome. The variables reviewed included a history of ocular trauma, contact with a vegetative foreign body, previous ocular surgery, recurrent herpetic keratitis, and contact lens use. A logistic regression model was utilized to assess the association between these risk factors and the likelihood of culture positivity.

All statistical analyses were performed using R, version 4.4.3 [24].

## 3. Results

Data were collected from 221 eyes (117 right eyes and 104 left eyes) belonging to 220 patients diagnosed with infectious keratitis of various causes. The cohort included 98 males and 123 females, with a mean age at diagnosis of 71.5 ± 3.5 years and a median age of 74 years. Most patients were of Caucasian ethnicity.

### 3.1. Etiology

A total of 141 cases (64%) were attributed to bacterial infections, based on clinical evaluation, empirical recognition, or cultural confirmation. Microbiological samples were obtained from 52 patients; 33 (63.4%) yielded positive cultures, enabling pathogen identification.

The most frequently isolated bacterial species were *Pseudomonas aeruginosa* (21 cases), *Klebsiella* spp. (5 cases), *Staphylococcus aureus* (4 cases), *Staphylococcus epidermidis* (2 cases), and *Serratia marcescens* (2 cases). Initial and targeted treatments primarily involved third- and fourth-generation fluoroquinolones (ciprofloxacin and moxifloxacin). In 52 cases (36%), fortified antibiotics, predominantly tobramycin (34/52), were added. The need for dual fortified therapy (e.g., cefazolin or vancomycin drops) was rare. Alternative antibiotics such as aminoglycosides or chloramphenicol were occasionally used in cases of intolerance to standard treatments. Systemic antibiotic therapy (ceftriaxone or ciprofloxacin) was required in 8 cases (5%), mostly in severe infections involving deeper ocular structures or progressing to endophthalmitis. Among these, 4 patients underwent rescue surgery. The mean duration of treatment for bacterial keratitis was 33 days. Rescue surgery to control infection was necessary in 8 cases (5%) (Table 1).

Fungal infections accounted for 29 cases (13%). Microbiological sampling was performed in 22 patients, with 15 (68%) resulting in positive cultures. The most frequently isolated fungi included *Candida* spp. (4 cases), *Aspergillus* spp. (3 cases), *Fusarium* spp. (3 cases), and *Alternaria* spp. (2 cases). Management involved topical antifungal therapy, mainly natamycin (24 cases), voriconazole (19 cases), or a combination of both agents (16 cases). Topical amphotericin B was used less frequently. Intrastromal injections of voriconazole (50 µg/0.1 mL) and amphotericin B (2.5 µg/0.1 mL) were employed in 13 cases to enhance infection control. Despite this, 6 patients ultimately required rescue penetrating keratoplasty (PKP). Notably, none of these patients experienced recurrence of fungal infection, a lower rate than typically reported in the literature (6–9%) [25]. Systemic antifungal therapy with oral voriconazole was initiated in 2 cases only.

The mean duration of treatment for fungal keratitis was 88 days. Rescue surgery was required in 11 cases (52.3%) (Table 2).

Seven cases (3%) were diagnosed with *Acanthamoeba keratitis*. All suspected cases underwent microbiological testing, achieving a 100% positivity rate. The primary treatment involved topical poly-hexamethylene biguanide (PHMB) 0.08% in all patients, with some also receiving combination therapy with chlorhexidine and hexamidine. Despite medical therapy, three patients (43%) required surgical intervention to control the infection. The average treatment duration for *Acanthamoeba keratitis* was notably long, at 235 days (Table 3).

Herpetic keratitis was diagnosed in 44 cases (20%). In 26 uncertain cases, microbiological samples were collected, PCR analysis successfully identified the causative viral agents, namely *Herpes simplex virus* (HSV) in 39 cases and *Herpes zoster virus* (HZV) in 5 cases. All viral keratitis cases were treated with topical antivirals (acyclovir or ganciclovir), and systemic antiviral therapy was administered in 32 of 44 patients. Notably, two cases of chronic recurrent herpetic keratitis progressed to corneal perforation or near-perforation, requiring surgical management. The average treatment duration (excluding long-term prophylaxis) was approximately 40 days (Table 4) (Figure 1).

### 3.2. Risk Factors

We aimed to analyze the risk factors associated with the onset of bacterial, fungal, and parasitic infections, as well as the likelihood of positive microbiological cultures in the presence of these factors. For this purpose, only the results of culture-based investigations were considered, excluding PCR analyses and, therefore, viral infections, as these risk factors are not linked to an increased risk of herpesvirus infections. Thus, a total of 92 patients out of 221 were selected for microbiological analysis. Among these, 55 samples yielded positive culture results, while 37 were negative. Of the contact lens (CL) wearers, who numbered 94 in total, 44 patients underwent microbiological investigation, leading to the identification of infectious agents in 31 cases (70%). Specifically, bacteria were isolated in 22 cases, including *Pseudomonas* spp. (*n* = 16) and *Klebsiella* spp. (*n* = 5), with 5 instances of polymicrobial infection. Fungi were detected in 3 cases (*Aspergillus* spp., *n* = 2; *Paecilomyces lilacinus*, *n* = 1), and amoebae were identified in 6 cases. Infections that remained unidentified, as well as those not submitted for microbiological testing, were most likely bacterial in origin, as inferred *ab juvantibus*. The mean duration of treatment for keratitis among CL wearers was 57 days (Table 5). A statistically significant association (*p* = 0.01) was found between contact lens use and culture positivity, where wearing contact lenses increased the odds of a positive microbiological result approximately threefold (exp{1.2319}).

Regarding ocular trauma, 17 patients reported a traumatic event prior to the onset of keratitis, including 9 cases involving a vegetative foreign body; samples were collected from 9 of these patients, yielding positive cultures in 8 cases (88%). Identified pathogens included bacteria in 3 cases (*Pseudomonas* spp., *Staphylococcus aureus*, *Staphylococcus epidermidis*) and fungi in 5 cases (*Aspergillus*, *Alternaria*, *Bipolaris*, *Scaedosporium*, *Candida*). The remaining infections, whether undetermined or not subjected to microbiological analysis, were presumed bacterial in origin, except for two cases considered mycotic keratitis. The mean treatment duration for keratitis related to ocular trauma was 32.5 days. A statistically significant association (*p* = 0.02) was observed between a history of ocular trauma and a positive microbiological result, with ocular trauma increasing the odds of culture positivity approximately thirteenfold (exp{2.4423}) (Table 6 and Table 7).

### 3.3. Group Comparison: Positive vs. Negative Microbiological Analysis

Our analysis then focused on comparing the clinical outcomes of the positive and negative cohorts to assess the impact and significance of prompt, thorough microbiological diagnosis. The primary endpoint examined was final visual acuity. Specifically, we analyzed the change in visual acuity from the time of diagnosis to the best visual acuity achieved after treatment. When comparing the outcomes between the two groups, patients with negative culture results (group 0) showed a mean improvement in BCVA of 0.208, while patients with positive culture results (group 1) demonstrated a greater mean improvement of 0.348. The difference in mean visual improvement between the groups was statistically significant (*p* = 0.019). These findings indicate that identifying the causative microorganism through culture is associated with a significantly greater improvement in visual acuity (Figure 2).

Subsequently, we evaluate the need for rescue surgery, defined as a procedure with two main aims: to better control the spread of the infection by physically removing part of the infectious core and to attempt to preserve at least part of the integrity of the ocular structures. Nineteen patients underwent rescue surgery: 11 excisional keratoplasties (PK), 8 conjunctival flaps, 3 vitrectomies, and 1 enucleation. The keratitis results from bacterial causes in 7 cases, mycotic in 7 cases, amoebic in 3 cases, and herpetic in 2 cases.

All patients presented with an initial VA of 0.1 or less. Among those with negative culture results, approximately one in four (27%) required rescue surgical intervention. Conversely, among patients with positive culture results, only about one in seven (15%) needed this procedure. Therefore, patients with positive cultures appeared to have a lower need for rescue surgery compared to those with negative cultures, although this difference was not statistically significant (Table 8).

### 3.4. Group Comparison: Early and Late Referral Time

The subgroups of patients were then examined to evaluate possible relations between referral time and their clinical outcomes. The average referral time for bacterial keratitis (8.97 days) is notably shorter compared to that for fungal and amoebic keratitis (26.52 days and 28.66 days, respectively). Regarding herpetic keratitis, the sample shows considerable variability, as 50% of the cases were referred to our hospital within 10 days, while the remaining experienced longer waits, up to 60 days (Figure 3). The main visual outcome considered was the change in BCVA, representing the difference between final and initial BCVA. The analysis did not yield statistically significant results.

The need for rescue “à chaud” surgery was also examined in relation to the delay in referral to the Cornea ward. Among the patients who were admitted within 15 days of the keratitis diagnosis (*n* = 72), only 6 (8%) underwent a surgical procedure during the acute phase of treatment. In contrast, among patients whose access to the Cornea ward was delayed more than 15 days (*n* = 44), 18 (40%) required emergency surgery to control the disease. The analysis showed a strong and significant correlation between referral time and rescue surgery, with the likelihood of requiring surgery increasing by 2% for each additional day. Patients referred to our ward more than two weeks after their initial diagnosis had a longer treatment duration (mean 123 days) compared to those referred earlier than two weeks (mean 53 days), a difference confirmed by statistical correlation (*t*-test *p* < 0.001) (Figure 4).

## 4. Discussion

Microbial keratitis is a significant clinical condition as it can severely impair visual function and, in some cases, cause irreversible vision loss. Microbiological investigations are the gold standard for determining an etiological diagnosis, mainly to guide treatment and improve clinical outcomes. By identifying the causative microorganism and its antimicrobial susceptibility, microbiological testing helps clinicians tailor therapy more accurately, potentially shortening the infection duration, preventing structural damage to the cornea, and increasing chances of visual recovery [6]. Despite these benefits, microbiological diagnosis is not always routinely performed in all clinical settings, and its true impact on prognosis and the necessity for surgical intervention remains an ongoing area of research.

In our population of 221 eyes, clinical assessment combined with empirical therapy—often supported by microbiological analysis—revealed a predominance of bacterial keratitis (*n* = 141, 63.8%), followed by viral keratitis (*n* = 44, 19.9%), fungal keratitis (*n* = 29, 13,1%), and amoebic keratitis (*n* = 7, 3.2%). As a tertiary referral center for corneal pathology, we aimed to analyze the epidemiological characteristics of this patient cohort and to explore how microbiological investigations might influence the disease progression. In our setting, 48.4% of cases (107 out of 221 eyes) involved corneal sample collection to establish a precise etiological diagnosis. The overall culture positivity rate was 75.7% (81/107 patients), aligning with figures reported in other similarly designed studies [26,27]. Culture-based methods remain the gold standard for identifying most microbial pathogens, providing reliable sensitivity when corneal specimens are collected and processed correctly [6]. This highlights the importance of close collaboration between ophthalmologists and microbiology laboratories, as well as the value of a well-structured diagnostic workflow in the initial management of infectious keratitis. Furthermore, PCR-based molecular assays proved to be a helpful adjunct in investigating suspected amoebic and herpetic keratitis, as well as in cases presenting with atypical or overlapping clinical features [13,17,18].

In our study, the most frequently identified bacterial species were Gram-negative organisms, particularly *Pseudomonas aeruginosa* (21 cases) and *Klebsiella* spp. (5 cases), while Gram-positive bacteria, such as *Staphylococcus aureus* and *Staphylococcus epidermidis*, were less commonly detected. This distribution aligns with findings from other long-term reviews [6,28,29]. However, other groups investigating bacterial keratitis have reported a predominance of Gram-positive species, including coagulase-negative staphylococci, as described by Lamas-Francis et al. [30], Kase et al. [26], and Ting et al. [11], *Streptococcus pneumoniae* as reported by Madduri et al. [31], and *Staphylococcus epidermidis* as noted by Ung et al. [27]. This discrepancy may partly result from the fact that not all patients in our study underwent microbiological analysis. The decision to perform sampling was left to the clinician’s judgment, based on factors such as clinical severity, presence of risk factors, and response to empirical therapy, similar to the approach used in the study by Russello et al. [6], conducted in a comparable geographic setting and with a similar design. Consequently, some of our cohort did not contribute to the full etiological spectrum of pathogens.

Among fungal infections, the most frequently isolated organisms were *Candida* (4 cases), *Aspergillus* (3 cases), and *Fusarium* (3 cases). *Acanthamoeba* was identified as the causative agent in 7 cases of corneal infection. All viral keratitis cases were attributable to Herpesviridae, specifically *Herpes simplex virus* (39 cases) and *Varicella zoster virus* (5 cases). Overall, these findings are consistent with existing literature, confirming that fungal and amoebic keratitis represent a minority of infectious keratitis cases in developed countries [32,33]. The overall outcome was predictably more favorable for infections of bacterial and viral origin, while fungal and amoebic keratitis more often resulted in low visual acuity and necessitated rescue surgery to manage the infection, in accordance with existing evidence [13,17,18,27,34].

Although establishing a direct link between contact lens use or ocular trauma and the increased risk of developing corneal infection was not the primary aim of this study, some notable observations emerged when analyzing the overall data, particularly concerning these two well-known risk factors [11,13,26,35,36]. In our cohort, 42.5% of patients (94 out of 221) were contact lens wearers. Importantly, most keratitis cases within this subgroup were caused by *Pseudomonas aeruginosa*. Furthermore, all cases of polymicrobial keratitis and every instance of amoebic keratitis occurred in contact lens users, emphasizing their well-documented role as a significant risk factor. Our analysis indicates that contact lens wear was linked to a threefold increase in the likelihood of obtaining a positive microbiological culture, providing a valuable guide for clinicians managing infectious keratitis in patients with such a history [11,37]. Seventeen patients reported prior ocular trauma, another established predisposing factor, especially when involving exposure to vegetable or organic material. Consistent with the literature, most keratitis cases in this subgroup were fungal in origin [11,26,27,38]. In our findings, a history of ocular trauma was associated with a 17-fold increase in the chance of obtaining a positive microbiological result.

Additionally, our research focused on comparing the clinical outcomes of two groups of patients who underwent microbiological or molecular analysis (*n* = 107). Patients in whom a pathogen was identified (*n* = 81, 75.7%) demonstrated a better final visual acuity compared to those with negative microbiological results (*n* = 26, 24.3%). Moreover, the group with positive cultures required rescue surgery less frequently. In contrast, similarly designed studies conducted by Yarimada et al. [34] and Bhadange et al. [39] reported no significant difference in final best-corrected visual acuity between the culture-positive and culture-negative groups. Additionally, in those studies, the culture-positive group more often required rescue surgical intervention and/or evisceration. These contrasting findings may be partially explained by differences in the microbiological profiles of the two groups. In our study, culture-positive cases were often associated with more aggressive pathogens, particularly Gram-negative bacteria such as *Pseudomonas aeruginosa*, which are known for their rapid progression and tissue-destructive potential. This aligns with the observations reported by Bhadange et al. [39] and Yarimada et al. [34], who found that culture-positive infections were more frequently associated with severe complications and a higher need for surgical intervention. Conversely, the culture-negative group likely included infections caused by less aggressive or rapidly resolving pathogens, such as Gram-positive bacteria or viruses, which may respond well to empirical treatment. Furthermore, it is important to consider that culture negativity does not equate to diagnostic failure, as prior antibiotic use, low microbial load, or fastidious organisms may contribute to negative results despite ongoing infection. These factors may help explain why, in our cohort, differences in visual outcomes between the two groups, although statistically significant, were not as dramatic as one might expect.

In addition to microbiological factors, the timing of referral appears to be crucial. In many of our culture-negative cases, patients were referred to our center only after receiving empirical treatment in peripheral settings, often following a delay of more than 15 days. This prior exposure to topical or systemic antimicrobials may have decreased the effectiveness of microbiological investigations, and the absence of a confirmed cause may have contributed to a delay in starting targeted therapy—ultimately leading to worse clinical outcomes. Furthermore, in our cohort, patients with clinical signs suggestive of Gram-positive bacterial infections—typically associated with milder courses—were less often selected for microbiological sampling. As a result, this subgroup is likely underrepresented among culture-negative cases in our analysis, unlike in previous studies where such patients might have been included in the culture-negative group. This difference in selection criteria may also help explain why our culture-negative group showed relatively poorer clinical outcomes compared to those reported by Yarimada et al. [34] and Bhadange et al. [39].

Lastly, patients referred to our tertiary care center were stratified according to the interval between the initial diagnosis of keratitis and the baseline examination performed at our facility. Patients who presented within 15 days of their initial diagnosis experienced a shorter overall treatment duration, achieved higher final visual acuity, and demonstrated a lower need for rescue surgical interventions compared to those who were referred after more than 15 days. These findings suggest that early specialist evaluation and timely initiation of appropriate therapy play a critical role in improving outcomes in infectious keratitis. Furthermore, our analysis revealed that cases of bacterial keratitis tended to be referred to our center significantly earlier than infections caused by fungi or *Acanthamoeba*. This delay in presentation for fungal and amoebic keratitis may be explained by their often more insidious and less distinctive clinical manifestations, which can result in delayed recognition, prolonged empirical treatment in peripheral centers, and a consequent deterioration of the infection before specialist assessment is sought. Overall, these observations emphasize the importance of prompt referral and early microbiological and ophthalmological evaluation, particularly in cases with atypical or slowly progressive presentations, to optimize prognosis and reduce the risk of irreversible visual impairment.

This study presents some limitations that should be considered. Given its retrospective and observational design, the accuracy of certain data points—such as the exact onset of symptoms or prior treatments received before referral—may be subject to variability across clinical records. Additionally, the sample size for less common forms of keratitis, such as fungal and amoebic infections, was relatively small, which may limit the strength of conclusions drawn for these subgroups. Finally, as the research was conducted in a single tertiary care center, the findings may reflect specific referral patterns and clinical practices that are not necessarily generalizable to all ophthalmology settings.

## 5. Conclusions

Our study emphasizes the clinical importance of microbiological investigations in managing infectious keratitis, highlighting their association with better visual outcomes and a decreased need for rescue surgery. The findings emphasize the importance of early referral to specialized corneal services, as delayed access was significantly associated with longer treatment periods and higher rates of emergency surgical intervention. Additionally, the identification of contact lens use and ocular trauma as critical risk factors for culture positivity reinforces their role in guiding initial diagnostic decisions. Overall, our results support the incorporation of prompt microbiological testing and specialist assessment into the standard care pathway for patients with suspected infectious keratitis, aiming to improve prognosis and reduce vision-threatening complications.

## Figures and Tables

**Figure 1 microorganisms-13-02308-f001:**
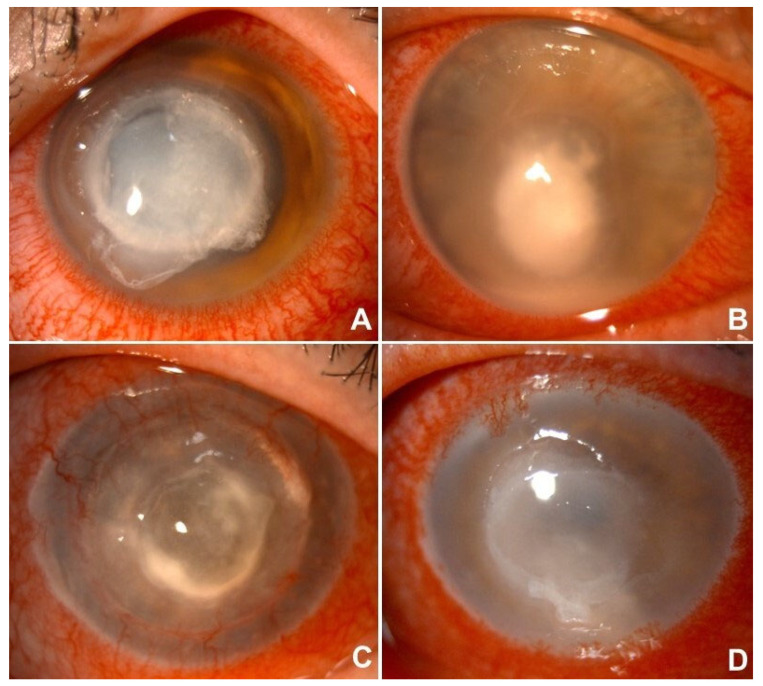
Anterior segment pictures of *Pseudomonas* keratitis (**A**), fungal keratitis (**B**), *Herpes simplex* keratitis (**C**) and *Ancanthamoeba* keratitis (**D**).

**Figure 2 microorganisms-13-02308-f002:**
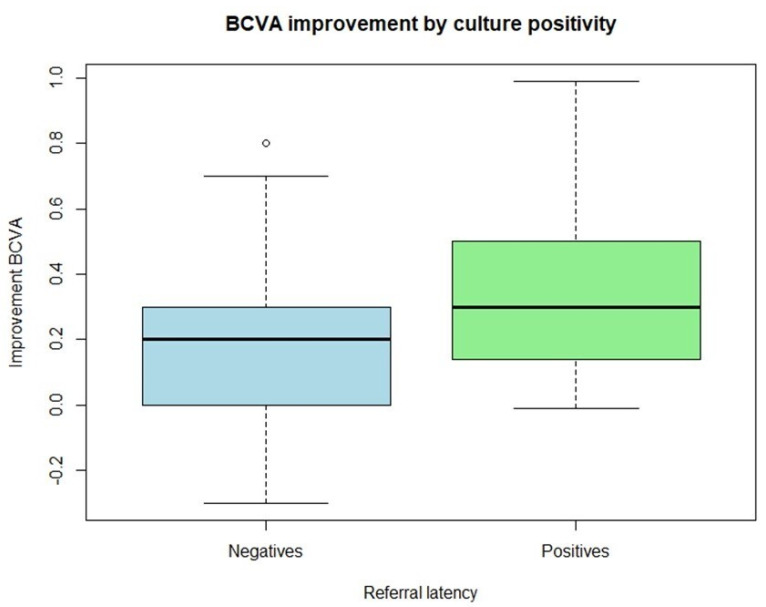
Patients with a positive microbiological culture showed greater improvement in BCVA compared to those with a negative culture.

**Figure 3 microorganisms-13-02308-f003:**
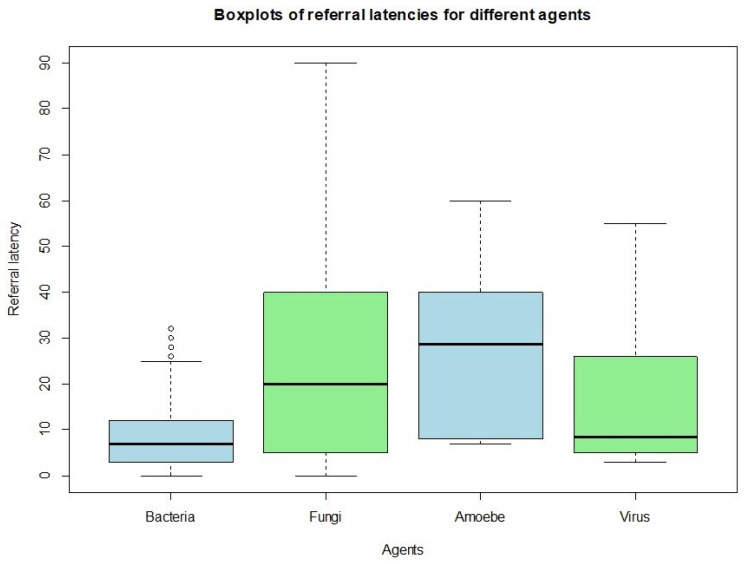
Distribution of referral times according to pathogen type, showing longer delays for fungal and amoebic infections compared to bacterial and viral infections.

**Figure 4 microorganisms-13-02308-f004:**
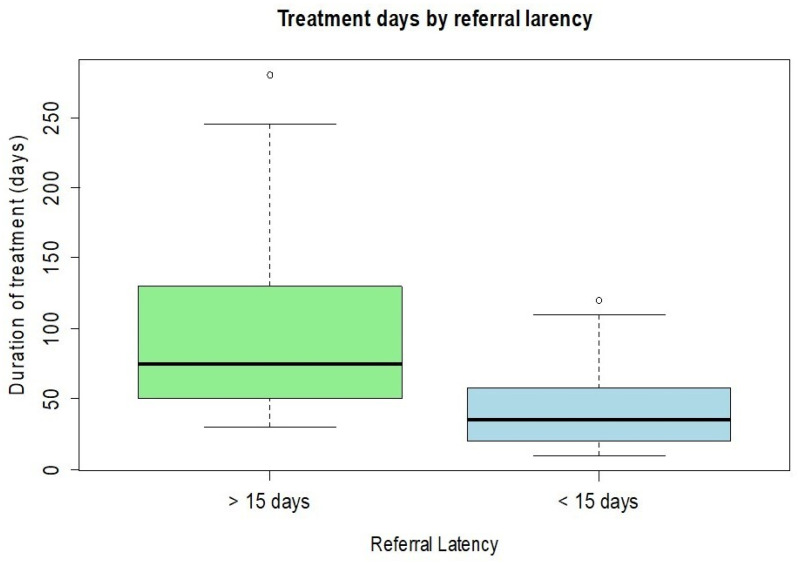
Patients with referral times >15 days required longer treatment durations compared to those referred within 15 days.

**Table 1 microorganisms-13-02308-t001:** Characteristics of bacterial keratitis.

Bacterial Keratitis	141 (64%)
Positive results at microbiology	33/52 (pos rate 63.4%)
- *Pseudomonas aeruginosa*	21 (40.3%)
- *Klebsiella* spp.	5 (9.6%)
- *Staphylococcus aureus*	4 (7.6%)
- *Staphylococcus epidermidis*	2 (3.8%)
- *Serratia marcescens*	2 (3.8%)
- Other	5 (9.6%)
- Polymicrobial infection	5 (9.6%)
Therapy	
- Fortified Antibiotics	52 (36.8%)
- Systemic Antibiotics	8 (5.6%)
- Rescue surgery	8 (5.6%)
Mean Duration of treatment	33 days

**Table 2 microorganisms-13-02308-t002:** Characteristics of fungal keratitis.

Fungal Keratitis	29 (13%)
Positive results at microbiology	15/22 (pos rate 68%)
- *Candida* spp.	4 (18.1%)
- *Aspergillus* spp.	3 (13.6%)
- *Fusarium* spp.	3 (13.6%)
- *Alternaria* spp.	2 (9%)
- Other	3 (13.6%)
Therapy	
- Topical Natamycin	24 (82.7%)
- Topical Voriconazole	19 (65.5%)
- Natamycin + Voriconazole	16 (55.1%)
- Intrastromal Injections (Voriconazole + Amphotericin B)	13 (44.8%)
- Systemic Antimycotic	2 (6.8%)
- Rescue Surgery	11 (37.9%)
Mean Duration of treatment	88 days

**Table 3 microorganisms-13-02308-t003:** Characteristics of amoebic keratitis.

Amoebic Keratitis	7 (3%)
Positive results at microbiology	7/7 (pos rate 100%)
Therapy	
- PHMB 0.08%	52 (36%)
- Rescue surgery	3 (43%)
Mean Duration of treatment	235 days

**Table 4 microorganisms-13-02308-t004:** Characteristics of herpetic keratitis.

Herpetic Keratitis	44 (20%)
Positive results at microbiology	44/44 (pos rate 100%)
- *Herpes Simplex Virus*	39 (88.6%)
- *Varicella Zoster Virus*	5 (11.4%)
Therapy	
- Topical Acyclovir/Gancyclovir	44 (100%)
- Systemic Antiviral	32 (72.7%)
- Rescue Surgery	2 (4.5%)
Mean Duration of treatment	40 days

**Table 5 microorganisms-13-02308-t005:** Characteristics of keratitis in contact lens wearers.

Contact Lens Wearers	94
CL wearers examined by microbiology	44/94
Positive Result at microbiology	31/44 (pos rate 70%)
- Bacterial Keratitis	23 (52.2%)
- Fungal Keratitis	3 (6.8%)
- Amoebic Keratitis	6 (13.6%)
- Polymicrobial infection	5 (11.3%)
Mean Duration of treatment	57 days

**Table 6 microorganisms-13-02308-t006:** Characteristics of keratitis in eye trauma.

Eye Trauma	17
Vegetable/Organic trauma	9
Patients examined by microbiology	9
Positive Result at microbiology	8/9 (pos rate 88%)
- Bacterial Keratitis	3 (37.5%)
- Fungal Keratitis	5 (62.5%)
Mean Duration of treatment	32.5 days

**Table 7 microorganisms-13-02308-t007:** Odds of culture positivity in contact lens wearers and in eye trauma.

	Estimate	*p*-Value
Intercept	−0.3629	0.2649
Trauma	2.4423	0.02771
Contact lenses	1.2319	0.0079

**Table 8 microorganisms-13-02308-t008:** Characteristics of recue surgery.

Rescue Surgery	19 (8.5%)
- Excisional Keratoplasty	11 (57.9%)
- Conjunctival Flap	8 (42.1%)
- Vitrectom	3 (15.8%)
- Enucleation	1 (5.2%)
**Etiological Agent**	
- Bacteria	7 (36.8%)
- Fungi	7 (36.8%)
- Amoebae	3 (15.8%)
- *Herpesviridae*	2 (10.6%)
Rescue surgery among positive microbiological analysis	12/81 (15%)
Rescue surgery among negative microbiological analysis	7/26 (27%)

## Data Availability

The raw data supporting the conclusions of this article will be made available by the authors on request.

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
