# Peer review of "Infectious Keratitis: A Tertiary Center’s Approach to Diagnosis, Management, and Enhanced Outcomes Through Microbiological Analysis"

_microorganisms, 2025, doi:10.3390/microorganisms13102308_

Round 1

Reviewer 1 Report

Comments and Suggestions for Authors

This manuscript conducted a retrospective review on 220 patients with infectious keratitis, with focus on the impact of microbiological analysis.

Only some minor concerns.

(1) For the case of microbial keratitis, the microbiological identification might be an important manner to determine an etiological diagnosis. In this case, the authors might highlight more on the finding of this manuscript, in comparison with previous reports.

(2) Accurate sampling of microbiological specimens in MK is an important step in identifying the infective organism. Corneal scrapping, tear samples and corneal biopsy are examples of specimens obtained for the investigative procedures in MK. Is it better to provide the sampling details in these patients?

(3) The application of different drugs or therapeutic approaches is due to the detection of bacterial species? If so, is it better to provide the bacterial characterization before, during, and after treatment?

Author Response

Dear Editors,

We thank you very much for your willingness to consider our manuscript mentioned above for publication in Microorganisms. We are therefore submitting a revised version of our article, which we hope will now be found suitable for publication.

Sincerely,

Antonio Moramarco MD

Reviewer 1:

This manuscript conducted a retrospective review on 220 patients with infectious keratitis, with focus on the impact of microbiological analysis.

Only some minor concerns.

  • For the case of microbial keratitis, the microbiological identification might be an important manner to determine an etiological diagnosis. In this case, the authors might highlight more on the finding of this manuscript, in comparison with previous reports.

Answer: Thank you for your observation. Microbiological analysis is increasingly crucial for the early detection of microbial agents in managing and treating infectious keratitis. While various groups have conducted similar studies, it is important to recognise that many environmental and sociodemographic factors may affect the results, making direct comparisons difficult. Notably, our study shares similarities in design, population, and area of investigation with the study by Russello et al., carried out in Reggio Emilia in 2021. Consequently, we focused our comparison mainly on this study. In the “Conclusions” section, we highlighted the importance of microbiological testing in diagnosing and treating microbial keratitis, which is the main focus of our research.

  • Accurate sampling of microbiological specimens in MK is an important step in identifying the infective organism. Corneal scrapping, tear samples and corneal biopsy are examples of specimens obtained for the investigative procedures in MK. Is it better to provide the sampling details in these patients?

Answer: We appreciate your suggestion. The methods employed to obtain the microbiological samples have been specified in lines 85–87.

  • The application of different drugs or therapeutic approaches is due to the detection of bacterial species? If so, is it better to provide the bacterial characterization before, during, and after treatment?

Answer: Thank you for the question. Etiological identification allows the use of the antibiogram to target the isolated species based on its results. If findings are negative, we use broad-spectrum antibiotics. We find it more useful to perform microbiological analysis only before starting treatment because conducting it during therapy is unlikely to yield valuable results and could even be misleading. The continuation of therapy depends on clinical signs, and we generally do not perform additional microbiological tests in the absence of evidence of infection.

Reviewer 2 Report

Comments and Suggestions for Authors

The submitted manuscript describes a clinical study on the prevalence of keratitis in a hospital. The authors present a clinical study on bacterial, fungal, viral und protozoal keratitis from one single hospital. The pathogens have been detected by standard microbiological methods. The data provided are not quite new or surprising. The discussion is adeqaute and cromprehensible. The detection method for Acanthamoeba should be mentioned.  

Minor critics:

line 85-90: how Acanthamoeba were detected? microscopy? or PCR

Table1: typing errors. The epitheton ornans of bacterial names should be given in small letters. (Pseudomonas aeruginosa etc.)Therapy: small letters or capiatl letters?? duration capital letters??

Table 2: capital letters? amphotericine B  without e.

 duration capital letters?

Table 4: mall or capital letters?

Table 5 small or capital letters? % : points or commas??

References. citation should be adapted to the standard requirements of the journal

ref 35: complete???

Author Response

Reviewer 2:

The submitted manuscript describes a clinical study on the prevalence of keratitis in a hospital. The authors present a clinical study on bacterial, fungal, viral und protozoal keratitis from one single hospital. The pathogens methods have detected the pathogens. The data provided are not quite new or surprising. The discussion is adeqaute and cromprehensible. The detection method for Acanthamoeba should be mentioned.  

Minor critics:

line 85-90: how Acanthamoeba were detected? microscopy? or PCR

      Answer: Thank you for the remark. Clarification has been added in lines 89-90.

Table1: typing errors. The epitheton ornans of bacterial names should be given in small letters. (Pseudomonas aeruginosa etc.) Therapy: small letters or capital letters?? duration capital letters??

      Answer: Thank you for the suggestion. Bacterial names have been corrected. The terms 'Therapy' and 'Duration' have been capitalised to indicate the respective categories.

Table 2: capital letters? amphotericine B  without e. duration capital letters?

      Answer: Thank you for your observation. The corrections have been made.

Table 4: mall or capital letters?

      Answer: The corrections have been made.

Table 5 small or capital letters? %: points or commas??

      Answer: The corrections have been made, and commas have been inserted where appropriate.

References. citation should be adapted to the standard requirements of the journal

ref 35: complete???

      Answer: Thank you for the observation. We have adapted the citations to meet the journal’s requirements and have completed reference 35.